# Sanitary Waters: Is It Worth Looking for Mycobacteria?

**DOI:** 10.3390/microorganisms12101953

**Published:** 2024-09-27

**Authors:** Angela Cannas, Francesco Messina, Paola Dal Monte, Francesco Bisognin, Giorgio Dirani, Silvia Zannoli, Giulia Gatti, Ornella Butera, Vincenzo Ferraro, Carla Nisii, Elena Vecchi, Giovanna Mattei, Giuseppe Diegoli, Antonio Santoro, Gian Luigi Belloli, Enrico Girardi, Tiziana Lazzarotto, Vittorio Sambri, Carla Fontana

**Affiliations:** 1National Institute for Infectious Diseases “Lazzaro Spallanzani” IRCCS, 00149 Rome, Italy; angela.cannas@inmi.it (A.C.); francesco.messina@inmi.it (F.M.); ornella.butera@inmi.it (O.B.); enrico.girardi@inmi.it (E.G.); carla.fontana@inmi.it (C.F.); 2Microbiology Unit, IRCCS Azienda Ospedaliero Universitaria di Bologna, 40138 Bologna, Italy; francesco.bisognin@studio.unibo.it (F.B.); vincenzo.ferraro3@studio.unibo.it (V.F.); tiziana.lazzarotto@unibo.it (T.L.); 3Department of Surgical & Medical Sciences-DIMEC, Alma Mater Studiorum-University of Bologna, 40138 Bologna, Italy; giulia.gatti12@unibo.it (G.G.);a.santoro.nefro@gmail.com (A.S.); vittorio.sambri@auslromagna.it (V.S.); 4Operative Unit of Microbiology, The Great Romagna Hub Laboratory, 47522 Pievesestina, Italy; giorgio.dirani@auslromagna.it (G.D.); silvia.zannoli@auslromagna.it (S.Z.); 5Collective Prevention and Public Health Sector, Directorate General for Personal Care, Health and Welfare, 40100 Bologna, Italy; elena.vecchi@regione.emilia-romagna.it (E.V.); giovanna.mattei@regione.emilia-romagna.it (G.M.); giuseppe.diegoli@regione.emilia-romagna.it (G.D.); gianluigi.belloli@regione.emilia-romagna.it (G.L.B.)

**Keywords:** dialysis water, *Mycobacterium saskatchewanense*, nontuberculous mycobacteria, whole-genome sequencing

## Abstract

The freshwater environment is suitable for nontuberculous mycobacteria (NTMs) growth. Their high adaptability represents a considerable risk for sanitary water systems, which are a potential vector for NTMs transmission. This study investigated the occurrence of NTMs, such as *Mycobacterium saskatchewanense*, in hospital water systems to support the surveillance and control of potentially pathogenic NTMs. We analyzed 722 ultrapure dialysis fluid samples from Emilia Romagna Dialysis Services. Among these, 35 samples were found to be positive for *M. saskatchewanense*. The strains were characterized using whole-genome sequencing (WGS) and variability analysis was carried out along the whole *M. saskatchewanense* genome. This investigation revealed the exclusive presence of *M. saskatchewanense* in these dialysis machines, with low genetic variability among all strains (with a low number of different alleles: <15). The strong similarity among the strain groups was also confirmed in the WGS-based ML tree, with very few significant nodes, and no clusters were identified. This research highlights the necessity of implementing surveillance protocols and investigating any potential link to human infections, as well as stressing the urgency of enhancing surveillance and infection control measures.

## 1. Introduction

Nontuberculous mycobacteria (NTMs) are a heterogeneous group of bacteria that are widespread and ubiquitous in the natural environment and can cause various infections in humans [1,2]. Although most NTMs species have not yet been associated with human diseases, the importance of NTMs infections has grown in recent years, increasing the challenge of treating clinical conditions, especially in immunocompromised patients [3,4].

In this context, the spread of NTMs represents a public health concern that requires rapid diagnosis and accurate surveillance programs to limit community and healthcare facility-associated infections [5]. Nevertheless, clusters of NTMs infections have been reported to be associated with comorbidities in patients, including individuals exposed to contaminated hospital potable water systems or those who are undergoing surgical interventions or other invasive procedures [6]. A well-known large-scale and multi-country outbreak of NTMs infection was caused by *Mycobacterium chimaera*, which is linked to contaminated heater–cooler units (HCUs). In 2014, several patients who underwent cardiac valve surgery developed endocarditis due to *M. chimaera* infection. Molecular epidemiological investigations identified the origin of the infection in contaminated HCUs used during surgical procedures within operating rooms. These devices allow *M. chimaera* transmission through the emission of aerosols that contain bacteria [7,8]. The sequencing and characterization of their whole genomes, as well as molecular epidemiological analysis, enabled the genotyping of *M. chimaera* strains, classifying them into genetic groups associated with the outbreak according to specific SNP profiles [9]. The *M. chimaera* outbreak affected patients from multiple countries, including Italy [10]. The event was closely monitored in Italy. An epidemiological study conducted by the Italian Ministry of Health across various regions identified 36 *M. chimaera* strains from heart surgery patients and 28 strains from HCUs, which were all linked to the outbreak both epidemiologically and genetically [11].

Moreover, the combined use of data concerning biochemical composition and genomic variability led to the characterization of 33 *M. chimaera* strains isolated from HCUs located in several facilities in Northern Italy. This identified the circulation of two groups of *M. chimaera* strains in the hospital water distribution system, resulting in different morphological and genetic types [12]. This global *M. chimaera* outbreak highlighted the need for enhanced surveillance and infection control measures for other NTMs in healthcare settings and international collaboration and coordination to identify and manage other similar events linked to NTMs [13].

In 2014, a correlation between *Mycobacterium abscessus* infections arising after lung transplantation or cardiac surgery and aerosol aspiration was hypothesized in hospitalized patients [14]. In 2021, whole-genome sequencing of two samples of *Mycobacterium chelonae* from pediatric hemato-oncology patients revealed high similarity to environmental samples from the hospital water supply [15]. Sequencing of the whole genome was indeed applied to define *Mycobacterium immunogenum* outbreaks between August 2013 and July 2021 [16].

NTMs can be difficult to treat due to their varying degrees of resistance to antibiotics. Treating NTMs infections often requires long and complex antibiotic regimens, which can lead to poor patient adherence and increased risk of side effects [17]. Some NTMs strains are also multidrug-resistant, requiring second-line or combination therapies for treatment [18]. Patient-related factors, such as comorbidities and immune status, can also impact treatment effectiveness [18,19]. Furthermore, a key challenge in treating NTMs infections is their ability to form biofilms, allowing them to evade the immune system and resist antibiotic treatment, leading to persistent infections [20,21]. Species like *Mycobacterium avium* and *M. abscessus* are known to form biofilms, contributing to their resistance to standard treatments [22]. The biofilm matrix acts as a barrier, preventing antibiotics from effectively reaching the bacteria [21]. Research has focused on finding new treatments to disrupt biofilms and improve the effectiveness of antibiotics, including the use of novel compounds such as thiopeptide antibiotics [20]. The ability of NTMs to form biofilms is a significant factor in their pathogenicity and resistance to treatment, as they have a selective advantage in contaminating medical devices and sanitary waters, posing a serious problem, as seen in *M. chimaera* [21]. Thus, despite recent findings indicating the importance of NTMs research, current monitoring and microbiological control procedures do not include NTMs surveillance [23].

Our study aimed to assess the presence of NTMs in hospital water sources, particularly in the water used during extracorporeal dialysis, and to explore their genomic diversity, providing considerable support for the surveillance of potentially pathogenic NTMs. We examined 722 ultrapure dialysis fluids collected from Emilia Romagna Dialysis Services in Italy and found that several samples were positive for *Mycobacterium saskatchewanense*.

## 2. Materials and Methods

### 2.1. Sample Collection

From August 2022 to August 2023, 722 ultrapure dialysis fluid samples were collected from all the Emilia Romagna Dialysis Services (*n* = 55), which belong to multiple producer companies. Furthermore, 10 L of sanitary water was collected from the distribution piping systems of nine dialysis services.

### 2.2. Sample Processing

Ultrapure dialysis fluid samples from dialysis machines were processed by the referral center for the detection of mycobacteria from environmental samples in the Emilia-Romagna Region, Italy, at the Microbiology Unit, IRCCS University Hospital of Bologna, and from the Microbiology Unit, The Great Romagna Hub Laboratory, Pievesestina.

Each water sample (1 L) was filtered with cellulose nitrate membranes (0.45 µm) using a Microsart filtration system (Sartorius, Göttingen, Germany) according to the ECDC guidelines, and the residue was resuspended in 10 mL of 0.9% saline solution [24]. Concentrated samples were treated as previously described [12]. Briefly, the samples were decontaminated with BBL MycoPrep solution (Becton Dickinson, NJ, USA), resuspended in 2 mL of phosphate-buffered solution, inoculated with Middlebrook 7H9 broth (MGIT, Becton Dickinson, USA), and cultured for 42 days. Ziehl–Neelsen staining was used to confirm the presence of mycobacteria in positive cultures.

*M. saskatchewanense* identification was achieved via matrix-assisted laser desorption ionization–time-of-flight mass spectrometry (MALDI-TOF) via a MALDI Biotyper^®^ system (Bruker, Bremen, Germany) from positive subcultures in Middlebrook 7H11 Agar (7H11 plate, Becton Dickinson, USA) [25]. The isolates were stored at –20 °C in Microbank™ cryovial (Pro-Lab Diagnostics, Richmond Hill, ON, Canada) and sent to the National Institute for Infectious Diseases “L. Spallanzani” (INMI) for molecular investigations.

### 2.3. Whole-Genome Sequencing and Bioinformatic Analysis

Thirty-five strains of *M. saskatchewanense*, which were isolated from sanitary water samples (ultrapure liquid) from dialysis machines, were shipped to INMI “L. Spallanzani” for molecular confirmation through WGS and bioinformatic analysis.

The genomic DNA of the strains was extracted from cultured MTBC isolates in liquid media (MGIT 960 or Middlebrook 7H9 Broth) using the QIAamp DNA minikit (Qiagen, Hilden, Germany) according to the manufacturer’s instructions. The quality and quantity of the extracted DNA were assessed using the Qubit Fluorometer (Thermo Fisher Scientific, Waltham, MA, USA) to ensure that the samples were of high-quality DNA suitable for subsequent analyses. The extracted genomic DNA was stored at –80 °C until further processing for library preparation and WGS. Before sequencing, the DNA was quantified using a Qubit 4.0 and a Qubit dsDNA HS assay kit (Thermo Fisher Scientific) and subsequently processed as described in the manufacturer’s protocol for WGS via the Nextera DNA Library Prep Kit and V2 or V3 flow cells on a MiSeq sequencer (Illumina, Inc., San Diego, CA, USA), with a pair-end approach. Genomic DNA was fragmented using enzymatic, followed by end-repair, adapter ligation, and PCR amplification. The resulting libraries were purified and quantified to ensure optimal library quality and concentration. Paired-end sequencing was conducted to generate short-read sequences with an average read length of 150 base pairs. The quality of the reads was evaluated via FastQC v0.12.1, and only the fastq files with high-quality standards were analyzed.

For the assembling step, SPAdes v3.15.5 was performed to reconstruct the bacterial genome into contigs, starting with trimmed reads, while CheckM2 v1.0.2 and Prokka v1.14.6 were used to obtain genomic characteristics and rapid bacterial genome annotation, respectively [26,27,28].

Sequence reads were first analyzed using SeqSphere+ software (RIDOM bioinformatics GmbH, Münster, Germany) (version 9.0.10 2023/9) for core genome multilocus sequence typing (cgMLST), represented by 5090 genes. The sequences were aligned with the reference genome (ID GenBank NZ_AP022573.1).

Additionally, Snippy version 4.3.6 (https://github.com/tseemann/snippy) (accessed on 26 July 2023) was used to perform mapping on the reference genome (ID GenBank NZ_AP022573.1) and the SNP variant calling. Concerning the procedures used to investigate whole-genome variability in this microbe, the following threshold variant parameters were used based on previous experience with *M. chimaera*: haploid genome, only SNPs, 5-fold minimum coverage, 90% allele frequency, 50 mapping quality, and 30 base quality [12]. IQTREE version 2.0, which is based only on SNP variability across the whole genome (~5.8 Mb), was used to create the phylogenetic maximum likelihood tree, and the significance of the nodes was assessed via bootstrap analysis with 10,000 replicates (>80% considered significant) [29]. The HKY + F + I model was selected as the best mutation model. The output was plotted using Figtree v1.4.3 (https://github.com/rambaut/figtree/releases) (accessed on 26 July 2023).

## 3. Results

### 3.1. M. saskatchewanense—Positive Samples

Positive *M. saskatchewanense* strains were isolated from 35 monitors/machines (4.6% of all samples) from nine dialysis services from PVS Forlì, Forli, Faenza, Santa Sofia, Savignano, Mercato Saraceno, Cesena, Ravenna, and Rimini. Positive *M. saskatchewanense* samples were collected from dialysis machines that used the chemical disinfection method (with peracetic acid). In contrast, when the same dialysis machines were disinfected via a citrothermic method (chemical and thermic), all samples collected (*n* = 420) were NTMs-negative.

Among the samples collected from the distribution piping system, *Mycobacterium lentiflavum* was detected only after double reverse osmosis at Cesena Hospital.

Lastly, WGS analysis was performed on 35 *M. saskatchewanense* strains isolated from pure water samples collected at several hospital sites in the Emilia Romagna region: 21 isolates were sent from the microbiology laboratory in Bologna (IRCCS Azienda Ospedaliero Universitaria di Bologna) and 14 were sent from Pievesestina (The Great Romagna Hub Laboratory).

### 3.2. Genomic Features of M. saskatchewanense Strains

The WGS analysis allowed the *M. saskatchewanense* species to be confirmed for all strains and a phylogenetic tree to be created. Raw sequence reads by WGS, as well as genome assemblies, are available under the BioProject accession number PRJNA1055936.

The genomic structure of each sequenced strain was investigated, providing the genomic assemblies and highlighting the genomic characteristics. Such data were reported in Appendix A. The assembling analysis highlighted an average genomic length of ~6.3 Mb, and 5935, as the average number of CDS. These values showed a higher structural diversity than *M. saskatchewanense* strain JCM 13,016 genome (NZ_AP022573.1), as it reported a genomic length of 6,008,916 bp.

### 3.3. Phylogenetic Analysis among M. saskatchewanense Isolates

To achieve a more conservative and comparable outcome, a core genome cgMLST panel consisting of 5090 genes was chosen based on annotations from NZ_AP022573.1, as these were the most informative regions. This selection helped us to construct a minimum spanning tree (MST) to determine the genetic distances between the strains (Figure 1). Details on the collected strains and the results of the WGS analysis performed with SeqSphere + RIDOM are shown in Table 1. Among these, 4377 genes coded for functional metabolic proteins and 128 coded for ribosomal proteins, while 585 were hypothetical proteins. The entire list of genes and metadata is reported in Appendix A (S_table_2_cgMLST_targets_5090). 

This explorative analysis showed that there was low variability among the *M. saskatchewanense* strains, as suggested by the low number of alleles per connection (<15). This condition could be attributed to high similarity among the *35 M. saskatchewanense* genomes, along with an absence of geographical correlation. Such a condition was also highlighted in ML tree-based genome sequencing, in which the strong distance from the reference genome (the only genome available in Genbank at the time of the study) and very few significant nodes confirmed the strong similarity with the strains group (Figure 2). The genomic variability analysis in *M. saskatchewanense* confirmed the species identification, and revealed low genomic diversity among the isolates, as shown by the cgMLST- and SNP-based phylogenetic analyses. Our results likely reflect the recent emergence of *M. saskatchewanense* as a distinct NTMs species detected in dialysis water systems.

## 4. Discussion

In this paper, we characterized the genomes of *M. saskatchewanense* strains isolated from water samples collected at several hospital sites, shedding light on their strong similarities in terms of the genomic structure of this potential emerging pathogen. Our results stressed the importance of investigating the genomic framework and the phylogenetic relationship among NTMs collected from hospital water systems.

NTMs are widely distributed in natural and man-made water sources, such as municipal tap water, hospital water systems, and dialysis fluids [25,30,31,32,33]. They exhibit a high degree of adaptation to their aquatic environments and have minimal survival requirements as free-living organisms in water. This ability allows them to endure in sterile deionized water for over a year. In their natural habitat, NTMs are typically found as free-living or biofilm-associated organisms that survive in lake water under hypoxic conditions [34,35].

Drawing from our previous experience with *M. chimaera*, our study aimed to analyze sanitary waters to detect the presence of NTMs. Results obtained from 722 samples obtained from nine distinct dialysis facilities revealed the exclusive presence of *M. saskatchewanense* in 35 machines. Whole-genome sequencing analysis demonstrated limited genetic variation across all strains, as opposed to the variability observed in *M. chimaera* [11,12]. Sparse information is available on *M. saskatchewanense*, which was initially identified in 2004 in a patient with recurring chest infections using a detailed examination of the 16S rRNA gene, ITS1 region, and hsp65 gene. Although this species was identified as closely related to *Mycobacterium interjectum*, the two mycobacteria species present many genetic differences [36]. In 2019, *M. saskatchewanense* was identified during a renal transplant evaluation in a patient with hereditary chronic renal disease that had required treatment with dialysis for the previous two years [30]. In 2023, this *Mycobacterium* species was isolated from ultrapure dialysis fluid samples from dialysis machines [25].

NTMs contamination of water can pose a serious infection risk for patients undergoing invasive procedures such as cardiac surgery, peritoneal dialysis, and organ transplantation, as well as for healthcare workers and visitors [9,13,25,35]. Using ultrapure sterile water is crucial for dialysis. During conventional hemodialysis, and even more so in methods like hemodiafiltration, approximately 200 L of water comes into direct or indirect contact with the patient’s blood. The presence of microorganisms, and even endotoxins, can cause severe septic states. Disinfecting dialysis systems (mainly water treatment plants and dialysis machines) is essential to ensure the safety and effectiveness of hemodialysis treatments. It is important to remove all microorganisms that are present, including NTMs. Disinfection methods include chemical and thermal disinfection, either used alone or in combination. Chemical disinfectants, such as citric or peracetic acid, are widely employed due to their broad antimicrobial efficacy. However, they may have limited effectiveness against certain resilient microorganisms, such as NTMs, which may exhibit inherent resistance or tolerance to chemical agents [37,38,39,40]. Indeed, the combined use of chemical and thermal methods, such as citrothermic disinfection, has shown greater efficacy than peracetic acid in combating NTMs and other waterborne pathogens [41]. Standards and procedures for dialysis water quality in clinical practice recommend daily disinfection of dialysis machines using bleach or heat/citric acid. Citric acid or acetic acid-based products can remove calcification or precipitate from within the internal hydraulics of the dialysis machine, while heat disinfection deletes pathogens and other types of microorganisms through thermal means [42,43,44,45]. Our results show that citrothermic treatment significantly improves the decontamination of *M. saskatchewanense* during the disinfection of dialysis machines and emphasize the need to monitor the quality and safety of dialysis fluids and the effectiveness of disinfection methods to prevent contamination and transmission of NTMs and other waterborne pathogens.

Although the environmental samples collected for this study may not fully represent the diversity of NTMs in dialysis fluids across different services or over time, the fact that *M. saskatchewanense* was the only species of mycobacteria found in dialysis machines is an interesting finding. Further studies should address the origin, evolution, biofilm-forming capacity, and genomic variability of this NTM species, as well as its potential pathogenicity and clinical relevance.

## 5. Conclusions

This study underscores the importance of continuous monitoring of dialysis water systems to improve our understanding and control of NTMs contamination in light of growing concerns caused by NTM disease associated with immunosuppression [23].

Because identifying the exact sources of NTMs contamination in dialysis systems remains challenging, and using water that is not ultrapure to make hemodialysis fluid might be a contributing risk factor, it is crucial to prioritize microbiological control and monitoring of the water used in this process. Specific disinfection methods should be routinely employed to prevent contamination by bacterial species that are resistant to chemical disinfectants.

In conclusion, considering the growing evidence of the ability of mycobacteria to survive in medical devices and the potential health risks they pose, screening healthcare water systems for the presence of these microorganisms should become routine practice if we aim to ensure a higher standard of safety and prevent infections.

## Figures and Tables

**Figure 1 microorganisms-12-01953-f001:**
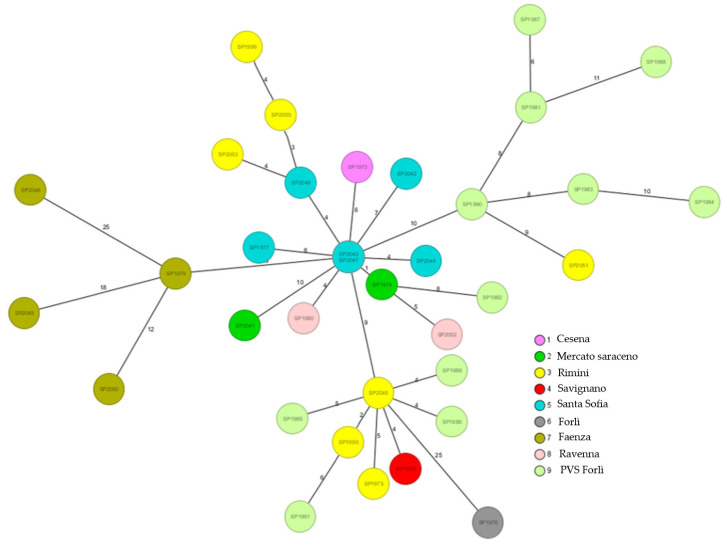
Minimum spanning tree of 35 *M. saskatchawanense* isolates based on cgMLST analysis using SeqSphere+ RIDOM (version 9.0.10 2023/9). software with the reference genome (ID GenBank NZ_AP022573.1). Each circle represents a strain, the position of which is based on the sequence analysis of 5090 target genes. The colors indicate the city where the strains were isolated, as reported in the legend. The numbers on the connecting lines illustrate the allelic distances between strains. Most of these differences are associated with <15 alleles.

**Figure 2 microorganisms-12-01953-f002:**
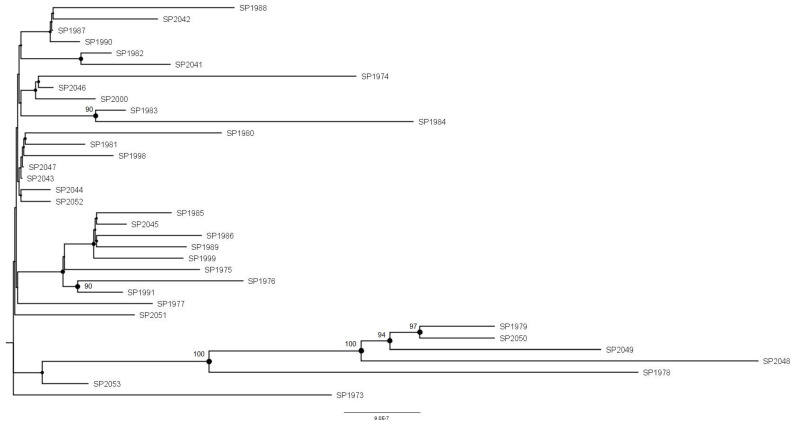
Unrooted phylogenetic ML tree based on the whole genomes of 35 *M. saskatchawanense* isolates collected in this study. The black dots are nodes with a significance >80%.

**Table 1 microorganisms-12-01953-t001:** Results of WGS analysis.

Strain ID	Sequence ID	Site of Collection	Date of Sample Collection	Species	% of Genome Coverage	Depth	Accession Number
1CE	SP1973	Cesena	25/10/2022	*M. saskatchewanense*	95.4	34	SRR27326215
1MS	SP1974	Mercato Saraceno	25/10/2022	*M. saskatchewanense*	95.2	29	SRR27326214
3RN	SP1975	Rimini	13/09/2022	*M. saskatchewanense*	95.6	38	SRR27326203
3SAV	SP1976	Savignano	20/10/2022	*M. saskatchewanense*	95.7	48	SRR27326192
5SS	SP1977	Santa Sofia	20/10/2022	*M. saskatchewanense*	95.6	42	SRR27326186
3OSM	SP1978	Forlì	08/02/2023	*M. saskatchewanense*	95.5	39	SRR27326185
12FAE	SP1979	Faenza	12/01/2023	*M. saskatchewanense*	95.7	42	SRR27326184
7RA	SP1980	Ravenna	02/11/2022	*M. saskatchewanense*	95.6	44	SRR27326183
MF4	SP1981	PVS Forlì	08/06/2022	*M. saskatchewanense*	95.3	59	SRR27326182
MF5	SP1982	PVS Forlì	08/06/2022	*M. saskatchewanense*	95.7	46	SRR27326181
MF7	SP1983	PVS Forlì	08/06/2022	*M. saskatchewanense*	95.7	45	SRR27326213
MF8	SP1984	PVS Forlì	08/06/2022	*M. saskatchewanense*	95.8	45	SRR27326212
MF9	SP1985	PVS Forlì	08/06/2022	*M. saskatchewanense*	95.7	50	SRR27326211
MF10	SP1986	PVS Forlì	08/06/2022	*M. saskatchewanense*	95.7	47	SRR27326210
MF12	SP1987	PVS Forlì	08/06/2022	*M. saskatchewanense*	95.7	65	SRR27326209
MF13	SP1988	PVS Forlì	08/06/2022	*M. saskatchewanense*	95.5	60	SRR27326208
MF14	SP1989	PVS Forlì	08/06/2022	*M. saskatchewanense*	95.6	50	SRR27326207
MF15	SP1990	PVS Forlì	08/06/2022	*M. saskatchewanense*	95.7	63	SRR27326206
MF16	SP1991	PVS Forlì	08/06/2022	*M. saskatchewanense*	95.8	75	SRR27326205
13538 61439/04	SP1998	Rimini	21/07/2022–18/08/2022	*M. saskatchewanense*	95.6	69	SRR27326204
13537 61430/X1	SP1999	Rimini	21/07/2022–12/08/2022	*M. saskatchewanense*	95.7	63	SRR27326202
13539 61439/03	SP2000	Rimini	21/07/2022–18/08/2022	*M. saskatchewanense*	95.8	59	SRR27326201
1SS	SP2044	Santa Sofia	07/11/2022	*M. saskatchewanense*	95.7	58	SRR27326200
2SS	SP2046	Santa Sofia	25/10/2022	*M. saskatchewanense*	95.7	56	SRR27326199
3SS	SP2043	Santa Sofia	25/10/2022	*M. saskatchewanense*	95.5	58	SRR27326198
4SS	SP2042	Santa Sofia	25/10/2022	*M. saskatchewanense*	95.3	51	SRR27326197
6SS	SP2047	Santa Sofia	25/10/2022	*M. saskatchewanense*	95.6	52	SRR27326196
7FAE	SP2049	Faenza	23/01/2023	*M. saskatchewanense*	95.7	68	SRR27326195
8FAE	SP2050	Faenza	12/01/2023	*M. saskatchewanense*	95.8	59	SRR27326194
14FAE	SP2048	Faenza	31/01/2023	*M. saskatchewanense*	95.5	56	SRR27326193
9RA	SP2052	Ravenna	20/10/2022	*M. saskatchewanense*	95.5	41	SRR27326191
6RN	SP2051	Rimini	13/09/2022	*M. saskatchewanense*	95.6	38	SRR27326190
10RN	SP2053	Rimini	11/10/2022	*M. saskatchewanense*	95.7	49	SRR27326189
13RN	SP2045	Rimini	13/09/2022	*M. saskatchewanense*	95.7	54	SRR27326188
3MS	SP2041	Mercato Saraceno	20/10/2022	*M. saskatchewanense*	94.9	45	SRR27326187

## Data Availability

Data can be found in the Excel database created ad hoc, which is archived at the authors’ institution (INMI L. Spallanzani IRCCS, Rome, Italy). The whole-genome sequencing files were submitted to NCBI BioProject ID PRJNA1055936.

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
