# Peer review of "Sanitary Waters: Is It Worth Looking for Mycobacteria?"

_microorganisms, 2024, doi:10.3390/microorganisms12101953_

Round 1

Reviewer 1 Report

Comments and Suggestions for Authors

The authors present interesting information on the isolation of Mycobacteria from water. The presentation of methods and results is clear. Below are some minor comments for the Authors:

- line 38 - in the keywords please give the full species name

- introduction: please supplement with information on antibiotic sensitivity/resistance among NTM strains. Are there potential therapeutic difficulties among patients ? 

- line 157 - lack of numbering

Author Response

The authors present interesting information on the isolation of Mycobacteria from water. The presentation of methods and results is clear. Below are some minor comments for the Authors:

We thank the reviewer for his valuable comments

- line 38 - in the keywords please give the full species name

Done

- introduction: please supplement with information on antibiotic sensitivity/resistance among NTM strains. Are there potential therapeutic difficulties among patients ? 

Additional details regarding the treatment of NTM have been incorporated. Given that our study does not concentrate on the therapy for Mycobacterium saskatchewanense and there's a scarcity of pertinent information in existing literature, we have opted not to discuss it explicitly.

.- line 157 - lack of numbering

The document has been extensively reviewed, and the accuracy of the line numbering is now assured.

Reviewer 2 Report

Comments and Suggestions for Authors

The manuscript entitled “Sanitary waters: is it worth looking for Mycobacteria?» is devoted to an important problem nontuberculous mycobacteria (NTMs) transmission risk in sanitary water systems. This study investigated the occurrence and genomic diversity of NTMs, such as Mycobacterium saskatchewanense, in ultrapure dialysis fluid samples to support the surveillance and control of potentially pathogenic NTMs. An important aspect of the work is the significant number of analyzed samples (722) from various medical institutions, which allows the authors to make adequate and global conclusions. To my mind this manuscript is topical and corresponding to the aims and scopes of the “Microorganisms” journal. I am ready to recommend it for publication after correcting several comments.

1. L161-177 should be moved to the introduction,

2. 177-187 Disinfection issues should be moved to the end of the discussion section

3. The discussion section should begin with an analysis of the data obtained in this work (L188)

4. It is worth adding a comparison of the data obtained by the authors with the results of other studies, most interestingly with data obtained in other countries, in order to understand how acute this problem is throughout the world.

5. The work lacks a conclusion. Perhaps part of the text on disinfection should be moved there.

Author Response

Reviewer 2

The manuscript entitled “Sanitary waters: is it worth looking for Mycobacteria?» is devoted to an important problem nontuberculous mycobacteria (NTMs) transmission risk in sanitary water systems. This study investigated the occurrence and genomic diversity of NTMs, such as Mycobacterium saskatchewanense, in ultrapure dialysis fluid samples to support the surveillance and control of potentially pathogenic NTMs. An important aspect of the work is the significant number of analyzed samples (722) from various medical institutions, which allows the authors to make adequate and global conclusions. To my mind this manuscript is topical and corresponding to the aims and scopes of the “Microorganisms” journal. I am ready to recommend it for publication after correcting several comments.

Answer: We would like to express our sincere gratitude to the reviewer for its valuable feedback.

  1. L161-177 should be moved to the introduction,

Answer We have completely revised the introduction as well as the discussion. We hope that the revised version fulfills your suggestions.

  1. 177-187 Disinfection issues should be moved to the end of the discussion section

Answer: We have completely revised the discussion. We hope that the revised version fulfils your suggestions.

  1. The discussion section should begin with an analysis of the data obtained in this work (L188)

Answer: We have completely revised the discussion. We followed your suggestion, and the text has been thoroughly edited.

  1. It is worth adding a comparison of the data obtained by the authors with the results of other studies, most interestingly with data obtained in other countries, in order to understand how acute this problem is throughout the world.

Answer: To the best of our knowledge, this manuscript presents the first comprehensive analysis of sanitary water that has identified the presence of Mycobacterium saskatchewanense. Consequently, comparison with prior studies poses challenges. Nevertheless, we have referenced pertinent literature, including the work by Turenne et al, which underscores the clinical significance of this microorganism, as well as the study by Di Mento et al, which similarly detected M. saskatchewanense in individuals with chronic kidney disease. Additionally, we have cited the research by Bisognin et al, a co-author of this work. While we can only postulate on the potential adverse impact of M. saskatchewanense in sanitary water based on the aforementioned studies, clinical data in this regard are currently unavailable.

  1. The work lacks a conclusion. Perhaps part of the text on disinfection should be moved there.

Answer: We thank the reviewer for their suggestion. We have added a conclusion section.

Reviewer 3 Report

Comments and Suggestions for Authors

See the attached PDF file.

Comments on the Quality of English Language

Moderate editing of English language required.

Author Response

The paper of Cannas et al., “Sanitary waters: is it worth looking for Mycobacteria” is devoted studying nontuberculous mycobacterium (NTMs). NTMs are a heterogeneous group of bacteria that are widespread and ubiquitous in the natural environment and can cause various infections in humans. Although most NTM species have not yet been associated with human diseases, the importance of NTM infections has increased, increasing the challenge of treating clinical conditions, especially in immunocompromised patients.

This is a quite interesting paper in general.

However, the obtained results are not clearly explained and justified, and not well-constructed. Results are given so generally and do not contain any outstanding novel scientific results. However, it is missing a lot of important details, and I will not recommend acceptance until these issues are addressed.

We appreciate the reviewer's insightful feedback.

1-Regarding topic of the paper. In the paper were discussed results of WGS of only Mycobacterium saskatchewanense strains. But the chosen topic is covered all NTMs? In my mind, the topic should be changed. Introduction

Answer: The purpose of this paper was to summarize the results of sanitary water surveillance focused on all NTM species. We were only able to identify that M. saskatchewanense was found in some samples tested as positive for NTMs. For this reason, we included the following sentence at the end of the introduction: "Our study aimed to assess the presence of NTMs in hospital water sources, particularly the water used during extracorporeal dialysis, and to explore their genomic diversity, providing considerable support for the surveillance of potentially pathogenic NTMs."

We have also added some sentences in the discussion and generally revised the entire paper.

2-The abstract is so general and does not contain any outstanding novel scientific results.

Answer: The abstract was rephrased.

3-I suggest, in introduction part, focusing on more M. saskatchewanense (their occurrence, ecology and physiology), than other spices of Mycobacterium. Because paper is about M. saskatchewanense. But of course, it will be given only one paragraph about other spices of Mycobacterium.

Answer: Thank you for the insightful feedback. We have completely revised the Introduction as well as the Discussion

4-There are insufficient of novelty and importance of this investigation. Authors should pay attention on what novel results did they get? The Results section is woefully inadequate.

Answer: The novelty of our work lies in the characterization of M. saskatchewanense strains from dialysis water systems for the first time, including the description of their phylogenetic relationship. Our results provide insights into a potential emerging pathogen, underscoring the importance of monitoring the quality and safety of dialysis fluids and the effectiveness of disinfection methods to prevent the contamination and transmission of NTMs and other waterborne pathogens. We have also included additional information in the Results section and have provided a supplementary material file.

5-Line 102. Positive M. saskatchewanense cultures were obtained from 35 monitor/machines. Did you isolate pure culture of M. saskatchewanense? It should be detailed.

Answer: every M. saskatchewanense strain was isolated from pure cultures, and this information has been added to the text.

6-Give some morphological and biochemical characters of isolated strains.

Answer: thank the reviewer for this suggestion. One M. saskatchewanense strain was already characterized morphologically and biochemically in our work by Bisognin and colleague (10.1038/s41598-023-48974-w). Furthermore, we reference the paper by Turenne et al, who were the first to describe such a microorganism.

7-Line 111. 2.2. M. saskatchewanense-positive WGS analysis. Did you perform shotgun metagenomic analysis of selected samples or you performed genome analysis of isolated pure cultures of M. saskatchewanense? Please, correct the name of this paragraph.

Answer:  We performed whole-genome sequencing analysis on M. saskatchewanense strains isolated from pure cultures. Consequently, the title of the paragraph seems appropriate. Adding more details to the methods section would likely make it clearer.

8-Line 117. Table 1. Results of WGS analysis. There is nothing to do with this table. There should be a table which presents genomic characterics of sequenced strains. And it should contain these: Completeness; Contamination; Coverage; Coding_Density; Contig_N50; Average_Gene_Length; Genome_Size; GC_Content; Total_Coding_Sequences; Total_Contigs; tRNA genes; Number of rRNA operons; Genome Accession number.

Answer:  In this table, for each sample we added genome accession number, as reported into the bioproject.

9- Move Fig.1 to the supplement materials or just exclude.

Answer:  We have excluded it. We agree that it did not add valuable information.

10-Authors performed WGS of 35 strains which is enough expensive analysis. However, the anticipated results from this analysis described and discussed so poorly. And I could not see why WGS was performed for these 35 strains. This paragraph should include minimum these subparagraphs: a) genomic features; b) Phylogenomic analysis (there also genomes have to be assigned taxonomy); c) Metabolic pathways (based on the annotation results; d) Pathogenic properties or something else (depends on what the authors wanted show).

Answer: I would like to express our gratitude to the reviewer for this valuable suggestion. Our manuscript focuses on molecular surveillance aimed at reducing waterborne disease transmission, improving sanitation, and controlling infections. Previously, we conducted a study on M. chimaera, as documented in our previous paper (10.1128/spectrum.02893-22).

It's important to note that sanitary water may be contaminated by nontuberculous mycobacteria, including M. chimaera, which is known to cause invasive infections in immunocompromised patients. By characterizing the genetic profile, we can gain insights into the origins of M. chimaera contamination. Our study applies this approach to investigate the potential emergence of this pathogen.

11-Line 123. Bioproject PRJNA1055936 is not found. Actually, the genome sequence of each strain should be deposited on NCBI Genome and should be obtained genome accession number. It only helps to check confidence of assembled genomes.

Answer: The text you provided seems to contain some errors. However, based on the content provided, a clearer version of the text could be:

Bioproject PRJNA1055936 has not been released yet. It will be released upon acceptance for publication.

Please find the link of bioproject: https://dataview.ncbi.nlm.nih.gov/object/PRJNA1055936?reviewer=53frnv23f3qkgmrpvb3gdnmvkd.

12-The results of cgMLST analysis did not open any biological pattern or described insufficiently.

Answer:  The results of cgMLST analysis were further described.

  1. Regarding phylogenetic tree. The quality of tree is bad. Is the tree was constructed based on concatenated alignments of 120 single-copy marker genes? Please, give the tree in the rectangular form, not as a circle form. Discussion

Answer: The phylogenetic tree was built on a multifasta file containing of whole genome consensus sequences of M. saskatchewanense (~5.8 Mb). This was the best representation of unrooted ML tree phylogenetic, but a rectangular version was updated.

14-Line 158-159. Authors reported that this study investigated the molecular epidemiology and genomic variability of M. saskatchewanense isolates. However, I do not see any outstanding results and discussion of these results. Molecular epidemiology is a branch of epidemiology and medical science that focuses on the contribution of potential genetic and environmental risk factors, identified at the molecular level, to the etiology, distribution and prevention of disease within families and across populations.

Answer: thank the reviewer for this suggestion. The term “molecular epidemiology” was deleted from the text.

15-There obtained results are not compared with other research. The discussion section is where you delve into the meaning, importance, and relevance of your results. It should focus on explaining and evaluating what you found, showing how it relates to your literature review and paper or dissertation topic, and making an argument in support of your overall conclusion.

Answer: We appreciate the reviewer's suggestion. We would like to provide more detailed commentary on the impact of M. saskatchewanense in healthcare settings, but unfortunately, there is limited literature available on this pathogen. Nonetheless, we have included some references and rephrased the discussion to align with the introduction, as per the request of reviewer 1.

16- Methods. How DNA was extracted exactly and with what kits. Did the authors do any controls? Negative DNA controls are important to detect kit/material contamination.

Answer: In this analysis, bacterial DNA was extracted without negative DNA controls, as each M. saskatchewanense strain was isolated from pure cultures. Additional information has been added in the methods, but we would like to emphasize that the majority of the methods are consistent with our previous publication (references now 11,12,25).

17-Details on the genomic sequence are completely lacking. This could be complicated (purification, library construction, adapters, what sequencing machine, paired-end reads?) and needs to be described in some detail.

Answer: Details about the genomic sequencing were added.

  1. The bioinformatics is inadequate. It is very important to tell the reader what specific database was used for taxonomic assignment of genomes? Was used GTDB database?

Answer: thank the reviewer for this suggestion. Snippy version 4.3.6 (https://github.com/tseemann/snippy) was used to mapping on the reference genome (ID GenBank NZ_AP022573.1), reported on NCBI, and the SNP variant calling steps. This bioinformatics approach is strongly conservative and currently used in the genomic investigation and bacterial typing, conducted by ECDC (www.ecdc.europa.eu) (i.e.  https://www.ecdc.europa.eu/sites/default/files/documents/Twelfth%20external%20quality%20assessment%20scheme%20for%20Salmonella%20typing.pdf)

  1. The genomes should be annotated and respectively all methods of annotation should be described.

Answer: The annotation was reported on reference genome ID GenBank NZ_AP022573.1.

Conclusions

20- Why authors did not give any conclusions? Conclusions have to be written. This section should not just be a summary; it should contain a take-home message.

Answer: thank the reviewer for this suggestion. The conclusion section was added.

21-Final comment: Dear authors, you have a huge genomic data set. However, you have not revealed the "essence" of this analysis and have not provided any new results on the molecular epidemiology of M. saskatchewanense. I suggest pay attention on more articles that adequately use the WGS results and then resubmit this manuscript, considering my comments

Answer: We have included additional information in the text and provided a supplementary file. We hope that this covers the "final comment" as well.

We thank a lot both reviewers for their precious suggestions and we tried to answer in detail, at point by point.

Round 2

Reviewer 3 Report

Comments and Suggestions for Authors

Dear authors, thank you for your responses to my comments. However, I am completely dissatisfied with your answers. Please do not avoid my comments and try to answer fully and accurately. 

Comment 8. Line 117. Table 1. Results of WGS analysis. There is nothing to do with this table. There should be a table which presents genomic characterics of sequenced strains. And it should contain these: Completeness; Contamination; Coverage; Coding_Density; Contig_N50; Average_Gene_Length; Genome_Size; GC_Content; Total_Coding_Sequences; Total_Contigs; tRNA genes; Number of rRNA operons; Genome Accession number.

Dear authors, you have added only the accession number of the raw WGS sequences to Table 1. I asked about the genome accession number of each genome assembly. This is a completely different number. To obtain these numbers, each assembled genome of your strain (there are 35 of them) must be deposited in NCBI Genomes. Genome Accession Number I need this to get more information about the genome assemblies of your 35 M. saskat. sp strains. Also, you have not provided the genome characteristics of your 35 strains. These are Completeness; Contamination; Coverage; Coding_Density; Contig_N50; Average_Gene_Length; Genome_Size; GC_Content; Total_Number_of_Coding_Sequences; Total_Number_of_Contigs; tRNA Genes; Number of rRNA Operons. If you are not able to perform these analyses, please involve a binoinformatician. I really need this information because I need to qualitatively evaluate your results.

Regarding 10 th of my comment: “Authors performed WGS of 35 strains which is enough expensive analysis. However, the anticipated results from this analysis described and discussed so poorly. And I could not see why WGS was performed for these 35 strains. This paragraph should include minimum these subparagraphs: a) genomic features; b) Phylogenomic analysis (there also genomes have to be assigned taxonomy); c) Metabolic pathways (based on the annotation results; d) Pathogenic properties or something else (depends on what the authors wanted show)”.

You answered about M. chimaera. Please , try to focus exactly on my comments. I am dissatisfied with your answer to my comment.

Comment 11. Line-123. I was clear there. I was asking about that the assemblies genome sequence of each strain should be deposited on NCBI Genome and should be obtained genome accession number. It only helps to me to check confidence of assembled genomes.

Comment 13. Regarding the phylogenetic tree. Thank you for the rectangular tree. Now I have a question: please note that SP1973 is on a completely different branch, and it seems that this strain may represent another species of mycobacteria. At least, the tree says so. That is why I asked above about the characteristics of the assembled genomes of your 35 strains. Please reclassify the assembled genomes of your 35 strains using the GTDB database. It seems to me that not all the genomes of the 35 strains are exactly Mycobacterium saskatchewanense. 

Comment 14. Line 158-159. I do not understand that why you deleted “molecular epidemiology”.  You have WGS results of 35 strains of M. saskat. sp and by this you can write surely about “molecular epidemiology”. 

Author Response

Comments and Suggestions for Authors

Dear authors, thank you for your responses to my comments. However, I am completely dissatisfied with your answers. Please do not avoid my comments and try to answer fully and accurately. 

Dear Referee,

Thank you for your feedback. We value your participation and the time you've dedicated to sharing your thoughts. We aim to address all comments comprehensively and precisely. If there are particular points you believe were not sufficiently covered, please understand that we have made every effort to provide a more detailed response.

Comment 8. Line 117. Table 1. Results of WGS analysis. There is nothing to do with this table. There should be a table which presents genomic characterics of sequenced strains. And it should contain these: Completeness; Contamination; Coverage; Coding_Density; Contig_N50; Average_Gene_Length; Genome_Size; GC_Content; Total_Coding_Sequences; Total_Contigs; tRNA genes; Number of rRNA operons; Genome Accession number.

Dear authors, you have added only the accession number of the raw WGS sequences to Table 1. I asked about the genome accession number of each genome assembly. This is a completely different number. To obtain these numbers, each assembled genome of your strain (there are 35 of them) must be deposited in NCBI Genomes. Genome Accession Number I need this to get more information about the genome assemblies of your 35 M. saskat. sp strains. Also, you have not provided the genome characteristics of your 35 strains. These are Completeness; Contamination; Coverage; Coding_Density; Contig_N50; Average_Gene_Length; Genome_Size; GC_Content; Total_Number_of_Coding_Sequences; Total_Number_of_Contigs; tRNA Genes; Number of rRNA Operons. If you are not able to perform these analyses, please involve a binoinformatician. I really need this information because I need to qualitatively evaluate your results.

Authors: the specific genomic characteristics of each sequenced strain were carried out in a supplementary table and cited in the main text as “S-table 1”.

Regarding 10 th of my comment: “Authors performed WGS of 35 strains which is enough expensive analysis. However, the anticipated results from this analysis described and discussed so poorly. And I could not see why WGS was performed for these 35 strains. This paragraph should include minimum these subparagraphs: a) genomic features; b) Phylogenomic analysis (there also genomes have to be assigned taxonomy); c) Metabolic pathways (based on the annotation results; d) Pathogenic properties or something else (depends on what the authors wanted show)”.

You answered about M. chimaera. Please, try to focus exactly on my comments. I am dissatisfied with your answer to my comment.

Authors. We are sorry for that. In the main text, we set the Results in two paragraphs: 2.2. Genomic features of M. saskatchewanense strains and 2.3. Phylogeentic relationship among M. saskatchewanense isolates, reporting in detail every information. As regard the metabolite pathways and pathogenic properties, we retain to not have enough information to discuss such aspects, but we guess it will be discussed in a next paper.

Comment 11. Line-123. I was clear there. I was asking about that the assemblies genome sequence of each strain should be deposited on NCBI Genome and should be obtained genome accession number. It only helps to me to check confidence of assembled genomes.

Authors. the assembled genome sequences were uploaded on NCBI Genome into the same Bioproject.

Comment 13. Regarding the phylogenetic tree. Thank you for the rectangular tree. Now I have a question: please note that SP1973 is on a completely different branch, and it seems that this strain may represent another species of mycobacteria. At least, the tree says so. That is why I asked above about the characteristics of the assembled genomes of your 35 strains. Please reclassify the assembled genomes of your 35 strains using the GTDB database. It seems to me that not all the genomes of the 35 strains are exactly Mycobacterium saskatchewanense. 

  • 35 strains were reclassify using the GTDB database. All genomes belonged to the M.XXX species.
  • The contigs of SP1973, as well as other 34 assembled genomes, were tested using BLASTn, showing to the Mycobacterium saskatchewanense species (AP022573.1), with more than 99% of identy.

Comment 14. Line 158-159. I do not understand that why you deleted “molecular epidemiology”.  You have WGS results of 35 strains of M. saskat. sp and by this you can write surely about “molecular epidemiology”. 

Authors: We considered that, in this case, the focus was not to reconstruct epidemiological dynamics, but to report presence of NTMs in hospital water sources, particularly in the water used during extracorporeal dialysis, and to explore their genomic diversity.

Thus, this term could be misleading and so we decide to delete it.